# Electroreductively Induced Radicals for Organic Synthesis

**DOI:** 10.3390/molecules28020857

**Published:** 2023-01-15

**Authors:** Huaming Xiang, Jinyu He, Weifeng Qian, Mingqiang Qiu, Hao Xu, Wenxi Duan, Yanyan Ouyang, Yanzhao Wang, Cuiju Zhu

**Affiliations:** Key Laboratory of Pesticides & Chemical Biology Ministry of Education, College of Chemistry, Central China Normal University, 152 Luoyu Road, Wuhan 430079, China

**Keywords:** electroreduction, radical chemistry, electrochemical synthesis, cathodic reduction

## Abstract

Organic electrochemistry has attracted tremendous interest within the novel sustainable methodologies that have not only reduced the undesired byproducts, but also utilized cleaner and renewable energy sources. Particularly, oxidative electrochemistry has gained major attention. On the contrary, reductive electrolysis remains an underexplored research direction. In this context, we discuss advances in transition-metal-free cathodically generated radicals for selective organic transformations since 2016. We highlight the electroreductive reaction of alkyl radicals, aryl radicals, acyl radicals, silyl radicals, fluorosulfonyl radicals and trifluoromethoxyl radicals.

## 1. Introduction

Radicals have played vital roles in drug discovery, agrochemicals, material science, and fine chemical manufacturing [1,2,3,4,5,6]. The current intense developments in radical chemistry particularly rely on the utilization of photoredox catalysis for the generation of radical species [7,8,9]. However, photoredox catalysts can absorb only a minor part of the solar spectrum (300–450 nm) [10]. As a complementary alternative, organic electrosynthesis has recently witnessed a remarkable renaissance for green and sustainable chemistry [11,12,13,14,15,16,17]. Meanwhile, electrochemically driven radical transformations have been attracting increasing interest in recent years [18,19,20,21,22]. The emergence of the first electro-induced radical processes can be traced back to the Kolbe electrochemical decarboxylation, which delivered dimerization of alkyl radicals by anodic oxidation [23]. In the 1960s, the first development of a cathodically generated radicals based method for synthesis of adiponitrile has been identified as the creation of an economically attractive approach for a scalable commodity chemical [24]. Compared with extensive studies on oxidative electrochemistry [25,26,27,28,29], reductive radical chemistry remains underdeveloped [30,31,32], due to several intrinsic challenges [33]. Firstly, the radical precursors usually have high reduction potentials (absolute value). Therefore, careful reaction design is often necessary to avoid the competitive side-reactions (e.g., protonation or substrate decomposed). In addition, the counter-anodic oxidation for cathodic reduction commonly employs the oxidation of a sacrificial anode wherein the anode is corroded, which has resulted in high current densities and irrelevant potential control. Despite these challenges, considerable advances have been made in cathodic electrosynthesis [34,35]. This review aims to highlight the key features of recent developments (2016–2022) in transition-metal-free electroreductive synthesis by cathodically generated radicals.

Radicals can be generated by cathodic reduction from a variety of radical precursors, such as alkyl or aryl halides, carbonyl compounds, and alkenes, among others (Figure 1A). The resulted radicals can be divided into three main categories of chemical transformations: (A) addition to a π-system by delivery of a new C-centered radical, which could be further reduced to carbanion, which then reacts with an electrophile; (B) a second cathodic reduction offers carbanion and then reacts with an electrophile directly; and (C) radical-radical cross-coupling occurent in persistent radical systems [36]. The reactivity and selectivity of cathodically generated radicals depends on their reduction potentials and relative rates (Figure 1B) [37,38].

## 2. Electrochemical Reduction

### 2.1. Electroreductive Reaction of Alkyl Radicals

Alkyl radicals are important synthetic intermediates that have widely contributed to the development of synthetic radical chemistry in recent decades [39]. The approaches for generating alkyl radicals are the reduction of alkyl halides, epoxides, cyclopropanes and ketones as well as iminium salts. The generated alkyl radicals could then proceed via two processes, in which (i) the alkyl radicals are further reduced to alkyl anions (a radical-polar crossover pathway) and then are trapped with the electrophile (E^+^) to deliver the functionalized alkylation products; and (ii) the alkenes act as radical acceptors to afford the new C-centered radicals, which are further reduced at the cathode to generate the corresponding carbanions and then react with another electrophile (E^+^) to construct the design products (Figure 2).

#### 2.1.1. Electroreductive Reaction of Alkyl Halides

Alkyl halides are the most simple and abundant reactants in organic synthesis and can be used as excellent alkyl radical precursors. They have frequently been engaged in novel organic reactions by their versatile reactivities and readily available properties. The high reduction potentials (absolute value) of alkyl halides (*E* = −1.5 to −2.5 V vs. SCE) are conceivably difficult to reduce to alkyl radicals [40].

Instead, in 2020, Lin group reported the electroreductive carbo-functionalization of alkenes with alkyl bromides via the radical-polarity crossover mechanism (Figure 3) [41]. Thus, value-added di-functionalization of alkenes was formed by the addition of two distinct electrophiles in a highly chemo- and regio-selective fashion. The new electroreductive strategies would be amenable to carboformylation, hydroalkylation, and carbocarboxylation of alkenes with a broad substrate scope and good functional group compatibility. The reaction started with the cathodic reduction of the alkyl bromide leading to an alkyl radical, which then was rapidly added to an alkene to give a new C-centered radical **6**. In the next step, the nascent C-centered radical **6** will be readily reduced to the carbanion intermediate **7**. Finally, the more inertly reductive electrophiles such as DMF (CHO donor), MeCN (H^+^ donor), and CO_2_ were suitable for the second C-C bond formation (Figure 3C).

Deuterated compounds have been widely applied in the realm of chemistry, mechanistic studies, and pharmaceutical science [42,43,44,45,46]. However, corresponding reports on unactivated alkyl halides have not been well elucidated [47,48]. Recently, the Qiu group showed electrochemically dehalogenative deuteration of un-activated alkyl halides (X = Cl, Br, I) or pseudo halides (X = OMs) using D_2_O as the economical deuterium source (Figure 4) [49]. This strategy is a good complement to the electrochemical deuteration of aryl halides, which has recently been successfully developed [50,51]. Here, a variety of natural products and pharmaceuticals were fully tolerated in the direct external catalysts-free reaction system. The proposed catalytic scenario consists of initial formation of the alkyl radical, which resulted from the cathodically reduced alkyl halides. Subsequent second electroreduction furnishes the alkyl carbanion intermediate **11** which then reacts with D_2_O and forms the final deuterated product. The *N, N*-diisopropylethylamine (DIPEA) combined with iodine anions (I^−^) serves as sacrificial reductants oxidized to iminium ion or iodine radical, respectively (Figure 4).

Although a few examples have been known for electroreductive or electro-photocatalytic reductive borylation of aryl halides [52,53]. The un-activated alkyl halides, especially bromides or chlorides, have rarely been explored for the electrochemical borylation. In 2021, Lu and co-workers first reported electroreductive transition-metal-free borylation of un-activated alkyl halides (X = I, Br, Cl) (Figure 5) [54]. The reaction proved compatible with primary, secondary, and tertiary alkyl halides to provide the desired boronic esters in a highly efficient manner. The mild electrochemical reaction conditions offered high chemoselectivities and various synthetically meaningful natural products and drug derivatives (Figure 5B). Based on detailed experimental and computational mechanistic investigations, a plausible mechanism was proposed for the electroreductive borylation of alkyl halides. It was thus suggested to initiate by cathodic reduction of B_2_cat_2_ the generation of B_2_cat_2_ radical anion species **25.** DFT studies showed that B_2_cat_2_ radical anion **25** both mediated the formation of the alkyl radical intermediate and trapped the alkyl radical, resulting in the formation of the final product **27** (Figure 5C).

The development of efficient C(sp^3^)-C(sp^3^) bonds construction has stimulated continued interest [55]. A wide variety of the couplings between an electrophilic organohalide and a nucleophilic organometallic agent, and a transition metal-catalyzed cross-coupling of two different carbon electrophiles have been reported [56,57,58]. These strategies, based on the prefunctionalized starting materials, underwent the undesired homocoupling side reactions. More recently, Lin described the electroreductive cross-electrophile coupling (XEC) [59,60] of the differential active alkyl halides driven by their disparate electronic and steric properties (Figure 6) [61]. The transition-metal-free methods provided high selectivity for the generation of a C-centered radical from the cathodic reduction of a more substituted alkyl halide, owing to its lower negative reduction potential (absolute value) compared to the less substituted alkyl halide (−2.13 V vs. −2.69 V, measured vs. Ag wire). Subsequent reduction of the resulting radical species **35** generated a carbanion **36**. This anionic nucleophile **36** was then subjected to react with various less substituted alkyl halide, thus completing the C(sp^3^)-C(sp^3^) bonds construction. The second reduction of radical intermediate **35** to the carbanion was difficult due to the facile proto-debromination and elimination side steps. It was critical to introduce an anion-stabilizing substituent to a more substituted alkyl halide, in order to reduce the potential for the second reduction event. Various functionalities substituents, such as boryl, aryl, vinyl, alkynyl and silyl, were compatible with the electroreductive conditions, providing the functional complexity and synthetic value of the cross-coupling products.

#### 2.1.2. Electroreductive Reaction of Epoxides and Cyclopropanes

In 2022, Lu and co-workers introduced an electroreductive methodology for regioselective hydrogenation of epoxides (Figure 7A) [62]. According to the density functional theory (DFT) calculations, anti-Markovnikov selective hydrogenation occurred preferentially with aryl-substituted epoxides, which originated from the thermodynamic stabilities of the in situ generated benzyl radicals **40a**. On the other hand, alkyl-substituted epoxides favored Markovnikov selective hydrogenation, which can be attributed to the kinetic preference for the formation intermediate **40b**. It was shown that the key to the success of this strategy was the implementation of utilizing TPPA ((pyrrolidino)phosphoramide), which had been applied as cosolvent and tripped the Mg anode as well as activated the epoxide. The proposed catalytic scenario consists of initially reductive epoxide to corresponding radical anion **39**. Subsequent rapid irreversible C−O bond cleavage of **39** furnished the C-centered radical intermediate **40a** or **40b**, which can further undergo electro-proton transfer to achieve the final products.

Concurrently, Yu reported the first electrochemically reductive dicarboxylation of small rings with CO_2_ (Figure 7B) [63]. Various valuable dicarboxylic acids were achieved efficiently by electroreduction of cyclopropanes and cyclobutanes with CO_2_. Moreover, different from electro-oxidative ring-opening difunctionalization, which are limited to the anodic oxidative radical cations’ intermediates [64,65,66]. The author first realized an electro-reductive ring-opening strategy, one which involved the key radical anions and carbanions intermediates. Under electroreduction, corresponding radical anion **42** was preferentially generated from single electron transfer (SET) reduction of the substituted small rings on the cathode, and then underwent nucleophilic attack on CO_2_ to give the carboxylated carbon radical **43**, which can be further SET reduced to yield C-centered anion **44**. Therefore, attacking another CO_2_ would generate the diacid products.

Subsequently, the Qiu group and the Zhang group independently developed the electroreduction of aryl epoxides with CO_2_ to afford valuable hydroxy acids with high selectivity and efficiency (Figure 7C) [67,68]. Moreover, three, four, five and six-membered aryl cyclic esters were suitable substrates, providing the corresponding β-hydroxy acids, γ-hydroxy acids and δ-hydroxy acids as well as ε-hydroxy acids, respectively. The proposed mechanism for these electroreductive carboxylations is depicted in Figure 7C. The formation of epoxide radical anion **46** was involved in two possible pathways in the initial step, which either underwent direct reduction on the cathode or was reduced by the generated CO_2_ radical anion. Then, intermediate **46** underwent C-O bond cleavage to give the C-centered radicals **47**. Subsequently, a further SET reduction of this C-centered radical **47** gave C-nucleophile species, and then the following attack of CO_2_ resulted in the formation of intermediate **48**. Finally work-up with aqueous HCl yielded the hydroxy acids product.

#### 2.1.3. Electroreductive Reaction of Ketones, Alkenes and Alkynes

As exemplified by the prior case study, the alkenes as the radical acceptors to be added with the cathodically generated radicals could be both a strategic and tactical assent for organic synthesis. Recent efforts by Baran and co-workers demonstrated that ketones and simple unactivated olefins were also viable substrates for access tertiary alcohols (Figure 8) [69]. The authors found that the sacrificial anode (Zn), electrolyte (*^n^*Bu_4_NBr) and current density all played critical roles to facilitate a high-yielding olefin-ketone coupling. The ketone **59** was initially reduced on the Sn cathode, whereupon reaction of the resulting ketyl radical anion **60** with alkenes gave rise to a C-centered radical **61.** The radical **61** can be reduced into the corresponding C-centered nucleophilic species **62** which can get a proton from α-position of ketone. Finally, the desired tertiary alcohols were achieved by workup (Figure 8C).

The hydrogenations of C-C multiple bonds as well as C-O double bonds are fundamental in organic synthesis. Traditionally, substrates are reduced with risky reagents (H_2_) [70,71], silicon hydrides [72,73], or hydrogen donors (formats, alcohols) [74,75,76]. However, safety hazards and undesired byproducts have compromised the principles of green chemistry. In 2019, Li and Cheng disclosed the electrochemical hydrogenation of alkenes, alkynes and ketones with gaseous ammonia as the proton source (Figure 9A) [77]. The use of ammonia was the key to ensure the lack of need of a sacrificial anode. The mild reaction conditions featured a broad tolerance of valuable functional groups, such as esters, amides and nitrile groups (Figure 9B). The selective hydrogenation of ketones, either to the corresponding alcohols or the diphenyl methane, relied on the applied voltage. The same group further expanded the scope of reductive electrosynthesis to the deuteration of α,β-unsaturated carbonyl compounds (Figure 9C) [78]. In this reaction, D_2_O severed as deuteration source as well as a sacrificial reductant avoiding cathodic corrosion. In addition, the detailed cyclic voltammetry investigations highlighted an initial reduction of the substrate **77** at −2.1 V vs. SCE, while the reduction potential of D_2_O (−2.5 V vs. SCE) was found to be higher (absolute value). These findings indicated that the substrate **77** had been reduced preferentially. The analogous electroreductive mechanism was depicted in Figure 9E. First, the alkene substrates **77** processed through SET cathodic reduction to form the anionic radical species **78**, which then coupled with a proton (deuton) transferring from ammonia or D_2_O to yield a radical **79** intermediate. The second cathodic reduction of resulting radical **79** generated the C-centered anion **80**. Finally, proton (deuton) transfer delivered the desired products **81a** or **81b**, respectively. Concurrently, N_2_ was generated from ammonia at the anode to fulfil the hydrogenation task in the first example. Notably, the ^18^O_2_ was captured with PPh_3_ to deliver the ^18^O-labelled triphenylphosphine oxide, which suggested the oxidation of D_2_O at the anode in the electrochemical deuteration reaction.

In 2019, another hydrogenation work was reported by the group of Xia using NH_4_Cl and methanol as hydrogen donors (Figure 10) [79]. Compared with previous hydrogenation of C=O double bonds in ketones [77], Xia reported the different selectivity hydrogenation of C=C double bonds in α,β-unsaturated ketones. The transformation proceeded through the aforementioned hydrogenation mechanism. Importantly, the authors reported that the solvent DMSO as the sacrificial reductant oxidized to methyl sulfone.

#### 2.1.4. Electroreductive Reaction of Cyanoheteroarenes

Reductions of cyanoheteroarenes have been well known to generate persistent radicals via photoinduced single-electron transformations [80,81,82]. A radical-radical coupling approach was proposed between this persistent and a transient radical. In 2020, Rovis and co-workers developed an electroreductive method for synthesis of the hindered primary amines (Figure 11A) [83]. Benchtop-stable iminium salts and cyanoheteroarenes were used as the radical precursors under mild reductive conditions. A proton-coupled electron-transfer (PCET) mechanism for the generation of radical **103** was supported by cyclic voltammetry studies and DFT calculations, as well as NMR spectroscopy. By the comparison of reduction potential of iminium salt **100** (−0.76 V vs. SCE) with cyanoheteroarene **102** (−1.3 V vs. SCE), the iminium salt **100** was preferentially reduced, furnishing α-amino radical **101**. The formation of **103** radical via the PCET pathway could undergo radical-radical coupling to forge the intermediate **104**. Finally, loss of H^+^ and CN^−^ would deliver the design hindered amine **105**. Building on this precedent, Xia group reported that the ketyl radicals, generated through reduction of aldehydes and ketones, can undergo facile radical-radical cross-coupling with the heteroaryl radical anion **106** providing secondary and tertiary alcohols (Figure 11C) [84]. The generated crucial aliphatic ketyl radicals were proved by electron paramagnetic resonance (EPR) studies. Concurrently, Jiang disclosed a mechanistically related transformation using thioester-activated alkenes as the radical precursors (Figure 12) [85]. This method was successfully applied to the synthesis of β-pyridine thioester. It should be mentioned that very recently Cheng group reported an electroreductively enantioselective Pd-catalyzed allylic 4-pyridinylation [86]. This method successfully employed the chiral η^3^ allyl Pd complex to capture the persistent radical via radical rebound.

### 2.2. Electroreductive Reaction of Aryl Radicals

Aryl halides served as essential reagents in organic chemistry and were common aryl radical precursors because of the inherent reactivity of aryl halogen (Ar-X) bonds. However, the high reduction potentials (absolute value) of aryl halides (*E* = −1.88 to −3.2 V vs. SCE), particularly the uncontrolled side reactions, compromised chemoselectivity, which remained a challenging area in electrochemistry. The general mechanism for the transformations of aryl radicals can proceed via three processes: (1) cathodic reduction to deliver the radicals anion, which falls in two categories: direct cathodic reduction or indirect cathodic reduction; (2) halogen heterolysis cleavage to form aryl radicals, and (3) the generated aryl radicals then proceed to the diversity of functionalized products (Figure 13).

#### 2.2.1. Direct Electroreductive Reaction of Aryl Radicals

The direct electroreductive system enables substrates to undergo electron transfer directly at the cathode surfaces. The direct approach results in the accumulation of highly reactive radical and radical anion intermediates in the electric double layer. Thus, these high-energy species can lead to further undesirable side reactions. Chi and co-workers reported a direct reductive progress to access hydro-dehalogenation of organic halides, which was significant for detoxification of environmentally hazardous organic halides (Figure 14A) [87]. The use of trialkylamines as suitable reductants and hydrogen atom donors to provide an electrochemical reductive system for replacing halogen atoms with hydrogen atoms. Later, the Zhang group proposed a procedure with copper nanowire arrays (Cu NWAs) as the recycled cathode to deliver the deutero-dehalogenation products (Figure 14B) [88]. The EPR measurements rationalized the existent aryl and hydrogen radicals. Moreover, the methods could be successfully applied to the paired electrolysis involving two desirable half-reactions. In 2021, Xia group disclosed the electroreductive methodology which was successfully extended to reactions other than hydro-dehalogenation, such as the reduction of Ts-protected amines and aromatic cyanides [89].

Due to the unique property of deuterated compounds, the development of user-friendly deuteron-dehalogenation has been reported by Lei and co-workers (Figure 14C) [51]. The commercially available electrodes, common battery and more user-friendly undivided cell highlight the potential practicability of this deuteration method. Initiation with cathodic reduction of aryl halides afforded the aryl radical (•Ar) 117 and bromine anion (Br^−^). D_2_O was reduced simultaneously to deliver the deuterium radical (•D), which can be trapped by the aryl radical to form the desired product 118 via radical-radical cross-coupling. The oxidation of bromine anion (Br^–^) at the anode followed by reaction with NPh_3_ delivered N(Ph)_2_PhBr.

The methodology for generating aryl radicals was not limited to the electroreduction of aryl halides, and electron-deficient arenes could be reduced at the cathode to generate the corresponding radical anions. Lei group developed the reductive radical cross-coupling of electro-deficient arenes and aryldiazonium (Figure 15) [90]. Various electro-deficient arenes and aryl diazonium tetrafluoroborate were employed to generate the corresponding arylation products. Moreover, anilines can be applied to synthesis diazonium reagents in situ, which can be further applied to the electrochemical arylation reaction. The highly effective nature of this reductive transformation is owed to the relatively low reduction potentials (absolute value) of quinoxaline (*E* = −1.06 V vs. Ag/Ag^+^) and aryl diazonium salt (*E* = −0.62 V vs. Ag/Ag^+^). It was also found that the EPR signals of quinoxaline radical anion and aryl radical were observed.

#### 2.2.2. Indirect Electroreductive Reaction of Aryl Radicals

In contrast, with indirect electroreductive reaction, a redox mediator, more easily reduced than the substrates, acts as the electron transfer shuttle from the heterogeneous electrode surface to the homogeneous dissolved substrates [91,92]. The indirect approach offers several advantages. In many cases, the introduction of a mediator results in improved reaction efficacy and better chemoselectivity by decreasing the overpotential required for substrate reduction. In 2016, Wan and coworkers showed that indirect cathodic reduction of arylhalides and pyrroles produced arylation of pyrroles using perylene disimide (PDI) as a mediator (Figure 16A) [93]. Mechanistically, aryl radical anion (Ar^•−^) **125** was produced through electro gain at the reduced PDI radical anion (PDI^•−^) species. The resulting (Ar^•−^) **125** underwent cleavage of C-X bond to deliver an aryl radical (•Ar) **126**. The pyrrole partner then reacted with aryl radical **126** to yield the arylation of pyrroles products. It has to be considered, though, that the radical anion (PDI^•−^) could further reduced to dianion (PDI^2−^). The radical anion (PDI^•−^) is proposed to be the more active catalyst species.

The electroreductive borylation of aryl iodides via aryl radical pathway was reported by the Mo group [54], which used pinacol diboron (B_2_pin_2_) as the borylating agent without the transition metal catalysts, reductants or sacrificial anodes (Figure 16B). The authors explored a series of mechanistic studies, such as EPR experiments and the cyclic voltammetry studies (CV), to reveal the electroreductive generation of aryl radicals.

In a subsequent report, Lin and Lambert jointly demonstrated electro-photocatalytic reductive transformation of aryl halides by using a dicyanoanthracene (DCA) as the electro-photocatalyst (Figure 17A) [53]. The transformation provided access to arylboronate, arylstannane and biaryl products by employing different radical trapping agents. The electro-photocatalysis relied on the electrochemical reduction of DCA to the corresponding radical anion (DCA^•−^) and followed by visible light photoexcitation to generate the stronger reductant [DCA^•−^]* radical anion (*E* = −3.2 V vs. SCE). The highly reducing photoexcited radical anion [DCA^•−^]* enabled facile reduction of aryl chlorides, which displayed high reduction potentials (absolute value) (*E* = −2.94 V vs. SCE) and strong bond dissociation energies (>97 kal/mol). Concurrently, a similar strategy was published by Wickens and co-workers who employed a naphthalene-based monoimide (NpMI) as electro-photocatalyst (Figure 17B) [94]. The resulting aryl radicals engaged with various radical traps to furnish phosphorylation and heteroarylation of aryl chloride. In this method, photoexcited reductant [NpMI^•−^]* has capabilities of reduction potential beyond those of Na metal.

The versatile electroreductive generation of aryl radicals were further applied to the intramolecular cyclisation of aryl halides. Recently, Brown group has showcased how the organic mediator-catalyzed electroreductive cyclisation reaction enabled the synthetically valuable five, six-membered cyclic or tricyclic ethers in flow (Figure 18) [95]. The flow reductive electrolysis was performed in efficient mass transport and narrow electrode area with the absence of a sacrificial anode. Sensitive functional groups cyano and 1-(allyloxy)-2-heterocyclic halides were thereby well tolerated. It was suggested to initiate by cathodic reduction of the phenanthrene mediator in order to deliver the strongly reducing phenanthrene radical anion. Then, electroreduction of aryl halides with the phenanthrene radical anion resulted in the formation of **148** intermediate, which undergoes the cleavage of halogen anion (X^−^) to form aryl radical intermediate **149**. The C-centered radical was trapped intramolecularly by a terminal alkene to form the five- or six-membered C-centered radical **150**. The latter C-centered was reduced by the second phenanthrene radical anion and protonation to give the final product **151**. The author unraveled the formation of hydrogenolysis byproduct by cathodic reduction **149** and then protonation in the absence of the phenanthrene mediator.

Very recently, Qiu group reported a mechanistically related carboxylation with CO_2_ as the C_1_ feedstock (Figure 19) [96]. Introducing the naphthalene as an efficient organic mediator led to unprecedented carboxylation of organic halides. A set of substituted aryl halides and heteroaryl halides as well as structurally complex drugs could be smoothly converted to the desired carboxylation products (Figure 19B). Based on detailed cyclic voltammetry (CV) studies, Qiu proposed a plausible reaction pathway involving the reductive mediator strategy. The relatively lower reduction potential (absolute value) of naphthalene (*E* = −3.09 V vs. Ag/Ag^+^) compared with aryl bromide (*E* = −3.53 V vs. Ag/Ag^+^) made it an available mediator to catalyze a wide array of organic halides. The cathodic reduction of naphthalene to form a naphthalene radical anion followed reduction with aryl halides forming intermediate **159**. The radical anion species **159** was dissociated to the active aryl radical (•Ar) **160**, which underwent cathodic reduction to give aryl anion (Ar^−^) **161**. Finally, nucleophilic attacking to CO_2_ afforded the design carboxylation product **162** (Figure 19C).

### 2.3. Electroreductive Reaction of Acyl Radicals

The electroreductive reactions were further extended to radical-radical cross-coupl- ing with benzoyl chloride and benzenesulfinic acid. Lei and co-workers recently demonstrated the synthesis of thioester derivatives via cathodic reductive acyl radicals and thiyl radicals in a user-friendly undivided cell (Figure 20) [97]. A wide range of substrates including heteroaromatic acyl chlorides, such as 2-furoyl chloride, 2-thiophenecarbonyl chloride and 2-chloronicotinyl chloride, were well tolerated. It is worthy of note that various alkyl acyl chlorides and alkyl sulfinic acid showed good compatibility in this electroreductive cross-coupling reaction (Figure 20B). Based on detailed mechanistic studies and the previous photochemical reductive reports [98], a plausible mechanism was proposed (Figure 20C). The reaction started with the cathodic reduction of Tf_2_O to generate [Tf_2_O]^•−^ radical anion and showed Tf_2_O mediate the formation of the acyl radical **172**. Therefore, the acyl radical **172** with benzenesulfinic acid furnished intermediate **173**, which was followed by cathodic reduction to an O-centered radical **174**. Trapping of **174** with acyl radical **172** resulted in the formation of the intermediate **175**, which underwent a second cathodic reduction to generate the thiyl radical **176**. Finally, radical-radical cross-coupling between acyl radical **172** and thiyl radical **176** led to the formation of the final thioester **177**.

### 2.4. Electroreductive Reaction of Silyl Radicals

In 2020, the group of Lin et al. reported an effective method to construct C-Si bonds for the synthetically valuable vicinal disilane by electroreduction of alkenes with commercially available chlorosilanes as the Si source (Figure 21) [99]. The protocol allowed for the construction of vicinal C-Si bonds of styrene, heterocycles and enynes. Moreover, a variety of chlorosilanes were efficiently transformed into the corresponding organosilanes (Figure 21B). The reaction started with the cathodic reduction of chlorosilanes to generate silyl radicals (•SiR_3_) in the presence of the anodic generation of magnesium salts (Mg^2+^). The producing •SiR_3_ sequential addition to the alkene provided the typically short-lived C-centered radical **187**, which can be further cathodically reduced to give **188** carbanion. Another electrophilic species, chlorosilanes, can readily undergo substitution to form designed disilylation products **189**. Notably, more challenging silacycles also proved viable substrates to deliver the corresponding five- and six-membered silacycles **190**. Additionally, a wide range of hydrosilylation products **191** were achieved by employing weakly acidic acetonitrile as the solvent.

### 2.5. Electroreductive Reaction of Fluorosulfonyl Radical

The photocatalyzed fluorosulfonylation has been achieved through sulfuryl chlorofluoride (FSO_2_Cl) as an effective fluorosulfonyl radical precursor [100,101]. Based on those results, the same research group also discovered an electrochemical synthesis of β-keto sulfonyl fluorides through reductive FSO_2_Cl with alkynes (Figure 22) [102]. The unstable FSO_2_ radical (•FSO_2_) could be generated via cathodic reduction of FSO_2_Cl. The alkene radical **196** underwent oxidation by O_2_ in the air to give the peroxy radical **197**, which then provided radical intermediate **198** following the Russell mechanism. Therefore, the resulting intermediate **198** can be the secondary cathodically reduced to deliver β-keto sulfonyl fluorides. Interestingly, α-chloro β-keto sulfonyl fluoride also can be achieved under THF as the solvent. The same synthetic strategy can also be applied to the fluorosulfonylation of vinyl triflates [103]. Note that this protocol avoided the use of a sacrificial anode and metal catalysis.

### 2.6. Electroreductive Reaction of OCF_3_ Radicals (•OCF_3_)

The tremendous progress have been made in electrochemical oxidative trifluoromethylation (CF_3_) of alkenes by CF_3_SO_2_Na as radical precursor [104]. Unfortunately, the electrochemical trifluoromethoxylation (CF_3_O) of (hetero)aromatics are inherently inefficient. Recently, the Qing group has gained significant momentum in this area by involving trifluoromethyl 2-pyridyl sulfone and oxygen as a trifluoromethoxylation source (Figure 23) [105]. As shown in Figure 23B, the mild electrochemical reaction conditions proved compatible with various synthetically meaningful functional groups, such as chloro, sulfonate, cyano on the (hetero)aromatics as well as bio-relevant molecules. The mechanism involved the formation of the •CF_3_, which was reduced from 2-pyridyl sulfone (**206**) with assistance of the hydrogen bond of HFIP solvent at the cathode. Subsequently, •CF_3_ combined with O_2_ to afford •OOCF_3_, which was secondarily reduced for the formation of the key •OCF_3_ intermediate. Next, the •OCF_3_ attacked the (hetero)aromatics to generate a transient C-centered radical species **207**, which was then oxidized at the anode and followed by rearomatization to provide the desired trifluoromethoxylated product **209**. Here, the additive Et_3_N served as the sacrificial reductant oxidized to the corresponding iminium cation.

## 3. Conclusions and Outlook

Electrochemical-reduction-induced radicals have been useful intermediates in organic chemistry. This Review has cited a large number of useful organic transformations, including alkenes functionalization, borylation, hydrogenation, silylation, arylation, dehalogenation and so on. Despite these advances, there are still many opportunities for the development of new and robust electrochemically reductive methods. The electroreductive reaction involves only one cathodic half-reaction, with the anode oxidation usually designed to be sacrificed. In order to achieve more efficient and selective molecular syntheses, electrochemical processes can be designed as a combination of two simultaneous desirable half-reactions, oxidation and reduction. Therefore, paired electrolysis can be considered as the gold standard for two desirable progressions on both electrodes [106,107]. On the other hand, there should be more advances made in asymmetric electrochemical radical reductive transformation [86,108]. We hope that this Review will inspire further developments in this exciting area of electrochemistry.

## Figures and Tables

**Figure 1 molecules-28-00857-f001:**
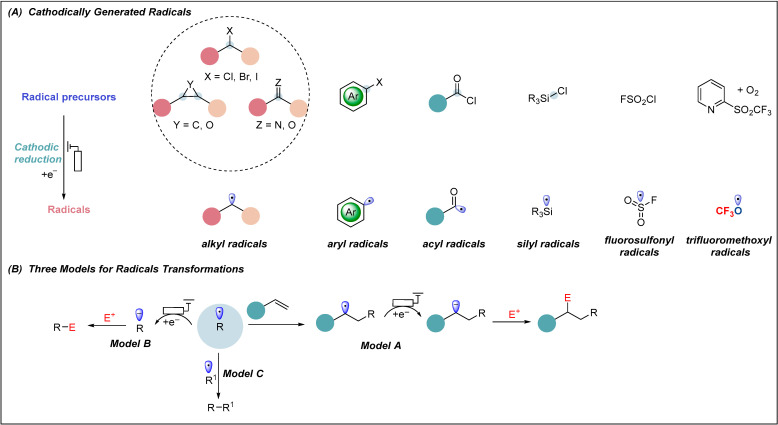
Cathodically generated radicals and the general mechanism of electroreductive transformations.

**Figure 2 molecules-28-00857-f002:**
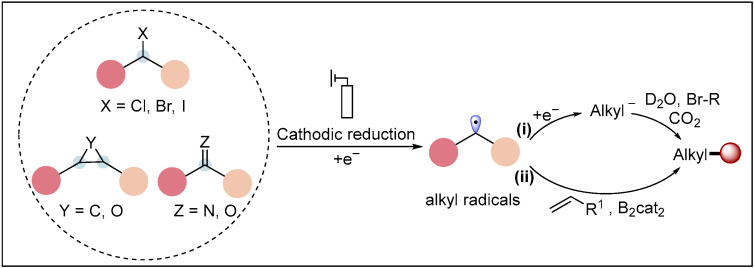
General mechanism of electroreductive transformations via alkyl radicals.

**Figure 3 molecules-28-00857-f003:**
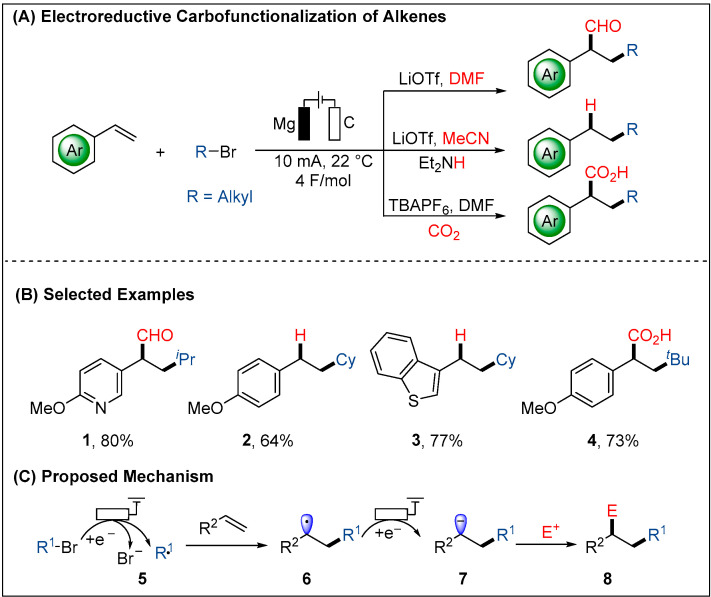
Electroreductive reaction of alkyl halides with alkenes.

**Figure 4 molecules-28-00857-f004:**
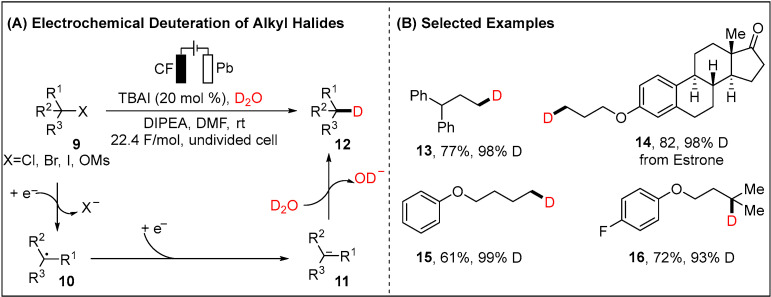
Electrochemical deuteration of unactivated alkyl halides by Qiu group.

**Figure 5 molecules-28-00857-f005:**
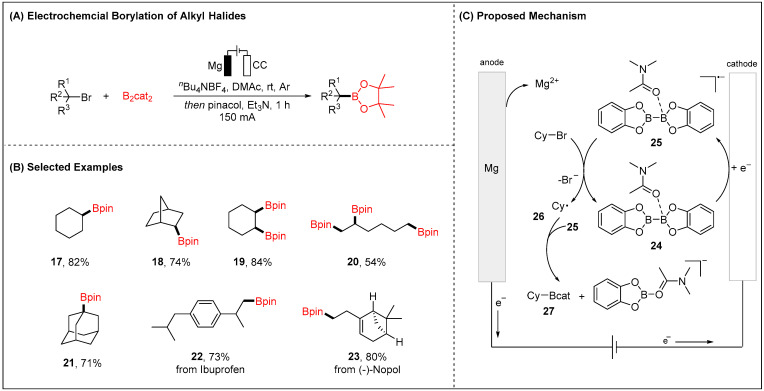
Electrochemical borylation of alkyl halides for alkyl boronic esters.

**Figure 6 molecules-28-00857-f006:**
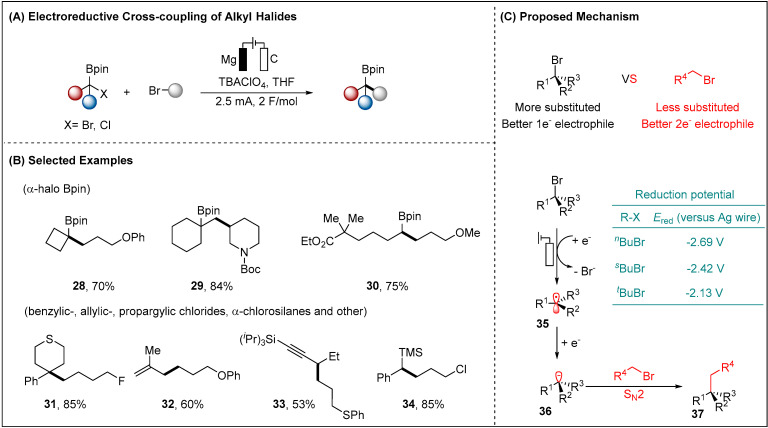
Electrochemically driven cross-electrophile coupling of alkyl halides.

**Figure 7 molecules-28-00857-f007:**
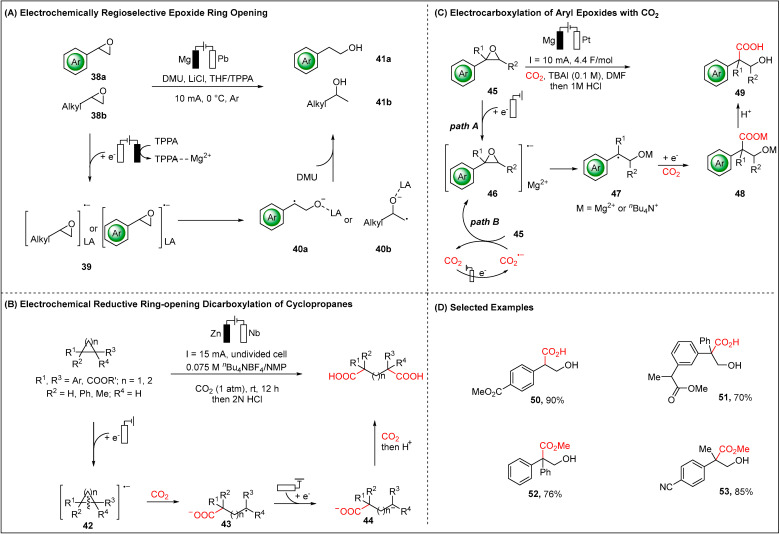
Electroreductionof epoxides or cyclopropanes.

**Figure 8 molecules-28-00857-f008:**
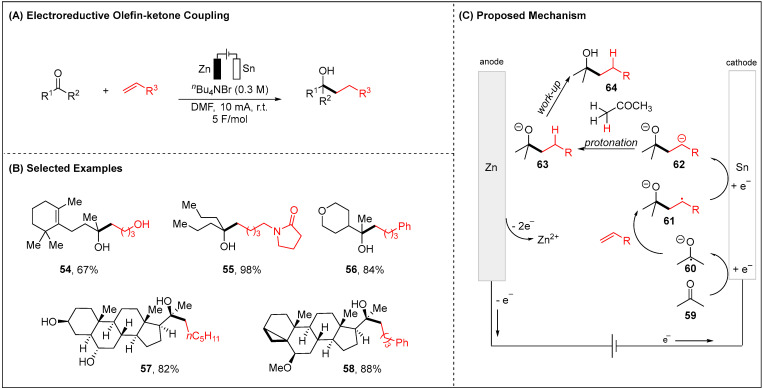
Electroreductive olefin−ketone coupling for the mild synthesis of tertiary alcohols.

**Figure 9 molecules-28-00857-f009:**
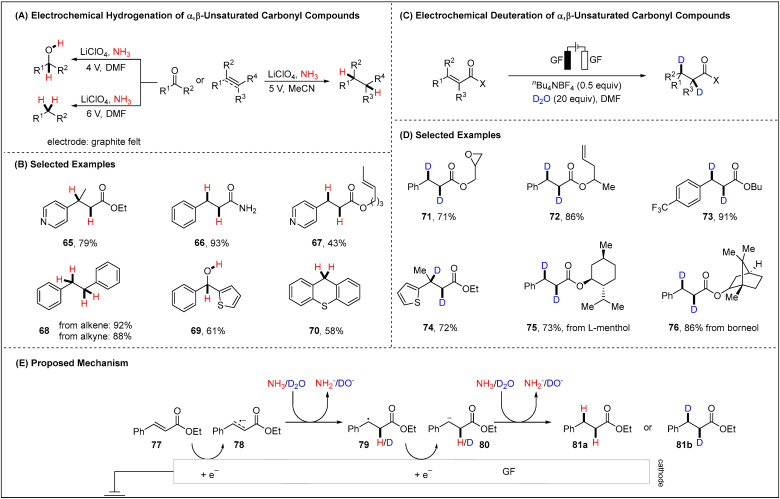
Electrochemical hydrogenation with gaseous ammonia or deuteration with D_2_O of alkenes, alkynes and ketones.

**Figure 10 molecules-28-00857-f010:**
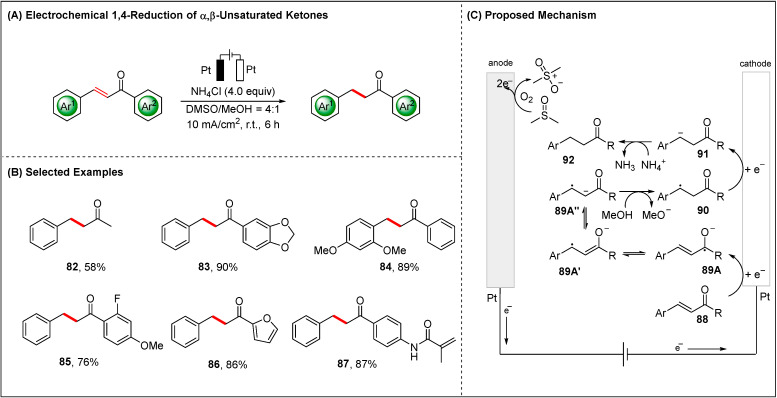
Electrochemical 1,4-reduction of α,β-unsaturated ketones with MeOH and NH_4_Cl as hydrogen sources.

**Figure 11 molecules-28-00857-f011:**
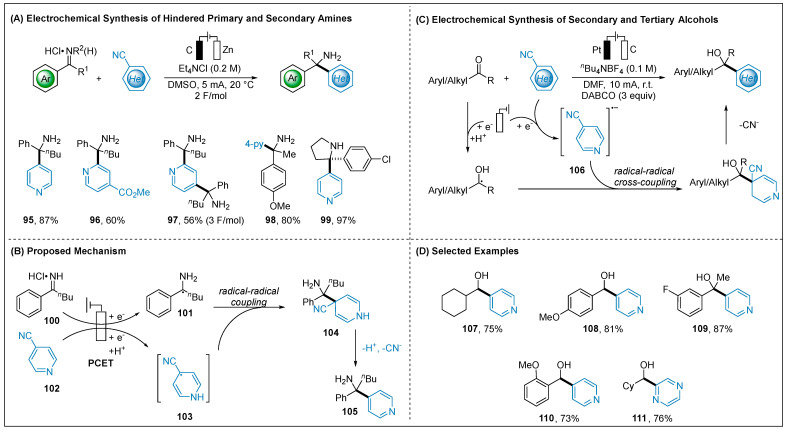
Electroreductive of cyanoheteroarenes with iminium salts or ketones.

**Figure 12 molecules-28-00857-f012:**
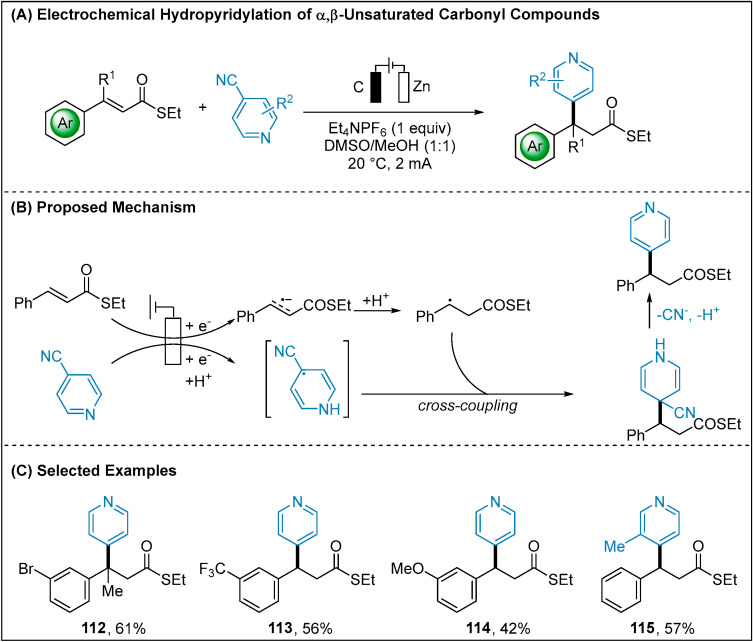
Electrochemical hydropyridylation of α,β-unsaturated carbonyl compounds.

**Figure 13 molecules-28-00857-f013:**
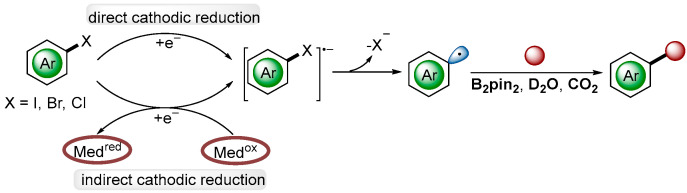
General mechanism for electroreductive transformation via aryl radical.

**Figure 14 molecules-28-00857-f014:**
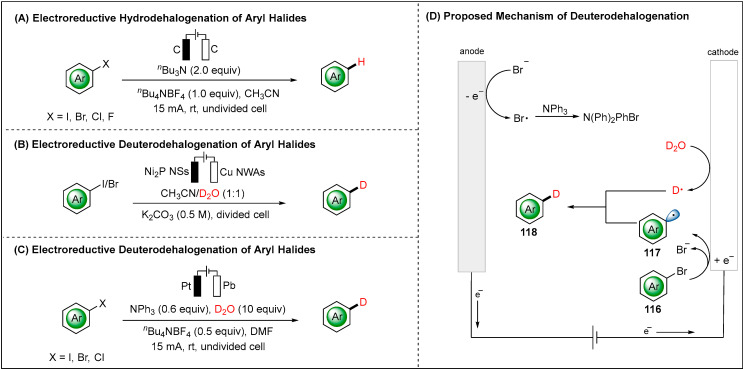
Direct electroreductive hydrodehalogenation and deuterodehalogenation of aryl halides.

**Figure 15 molecules-28-00857-f015:**
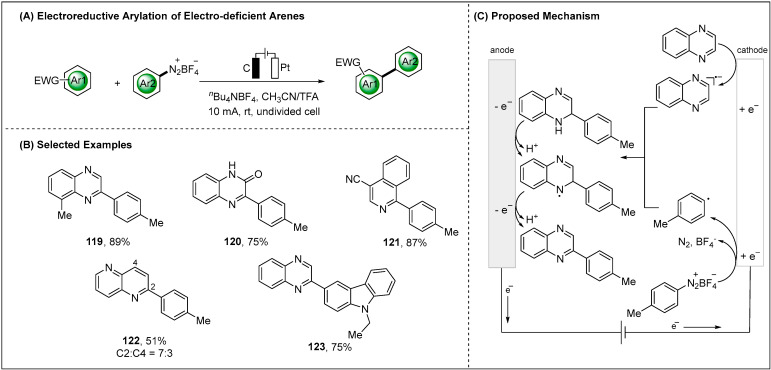
Direct electroreductive arylation of electro-deficient arenes.

**Figure 16 molecules-28-00857-f016:**
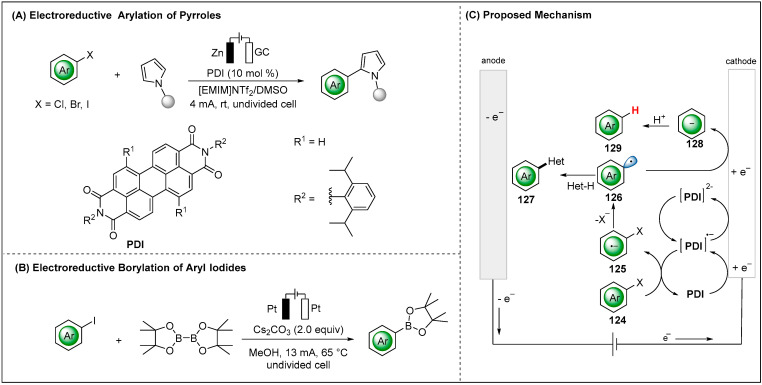
Electroreductive arylation and borylation of aryl halides.

**Figure 17 molecules-28-00857-f017:**
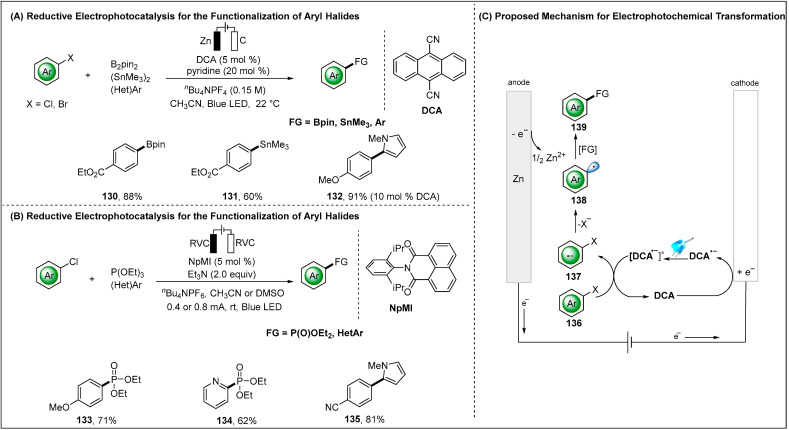
Electrophotochemical reductive transformation of aryl halides, DCA = dicyanoanthracene.

**Figure 18 molecules-28-00857-f018:**
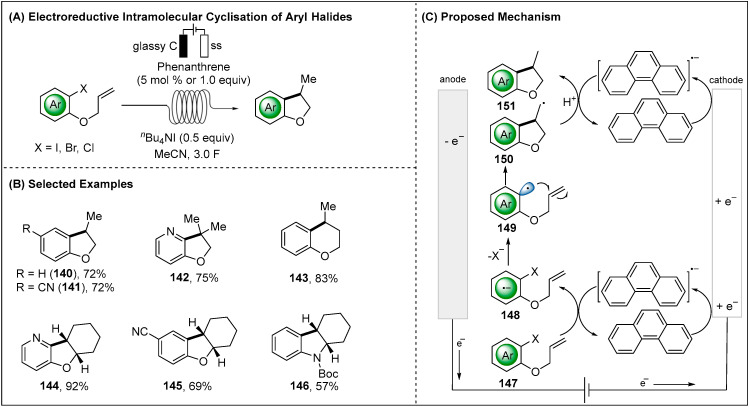
Phenanthrene mediated electroreductive intramolecular cyclisation of aryl halides by Brown group.

**Figure 19 molecules-28-00857-f019:**
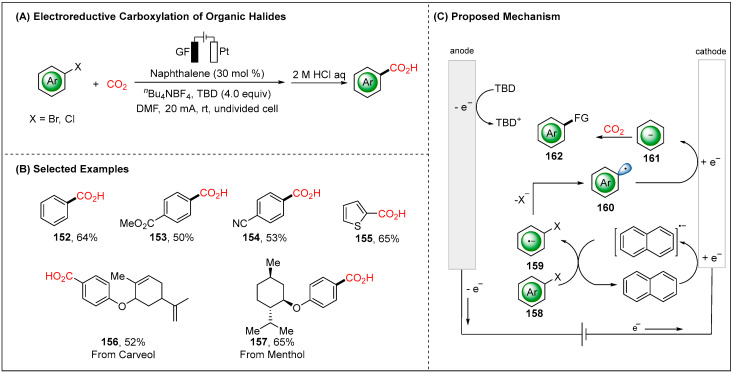
Naphthalene mediated electroreductive carboxylation of aryl halides by Qiu group.

**Figure 20 molecules-28-00857-f020:**
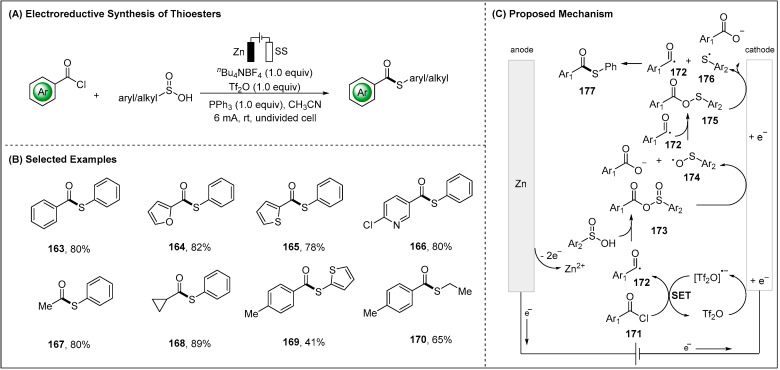
Electroreductive cross-coupling between acyl radicals and thiyl radicals by Lei group.

**Figure 21 molecules-28-00857-f021:**
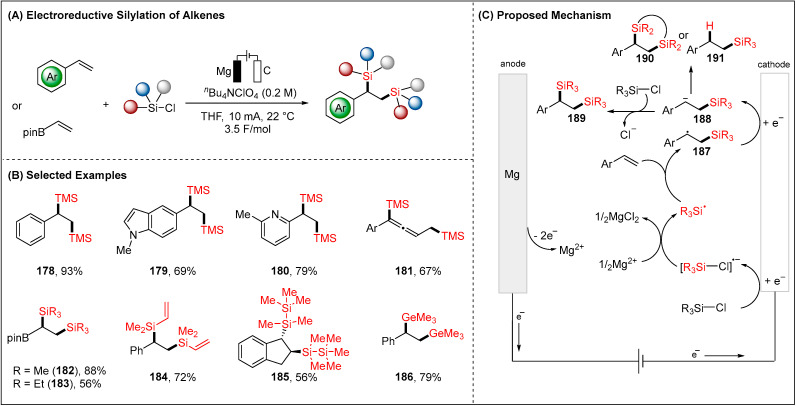
Electroreductive silylation of alkenes by Lin group.

**Figure 22 molecules-28-00857-f022:**
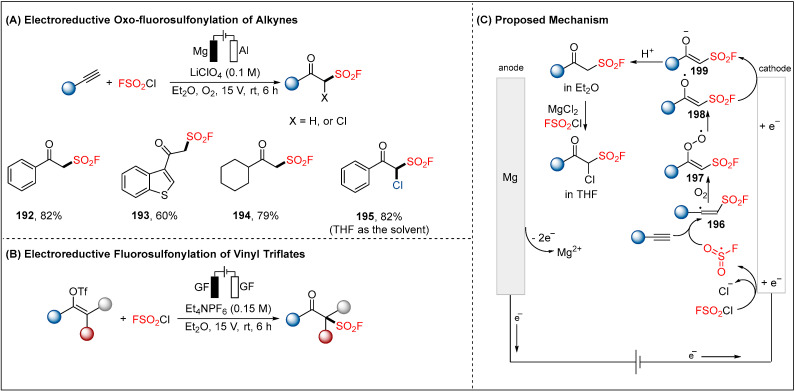
Electroreductive fluorosulfonylation of alkynes or vinyl triflates for the synthesis of β-keto sulfonyl fluorides by Liao group.

**Figure 23 molecules-28-00857-f023:**
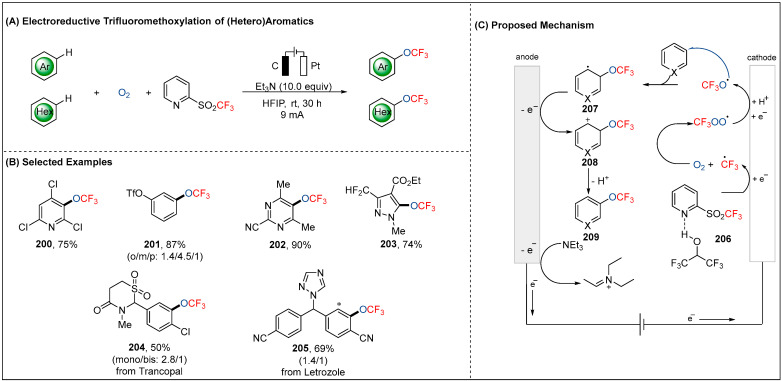
Electroreductive trifluoromethoxylation of (hetero)aromatics with a trifluoromethyl source and oxygen by Qing group.

## Data Availability

All data are available in the manuscript.

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
