# Peer review of "Electroreductively Induced Radicals for Organic Synthesis"

_molecules, 2023, doi:10.3390/molecules28020857_

Round 1

Reviewer 1 Report

Recommendation: Publish in Molecules after minor revisions noted.
Comments:

In this Review the authors discussed recent progress in the field of electrochemical reductive radical transformations. My impression of this Review is that the literatures covered here are presenting the progress of this field very well. For example, Lin, Qiu and Lei, among others, are active researcher in this field, have made significant contribution to cathodic reductive radical chemistry over past decade. The authors had shown various representative references to the important and elegant research efforts stemming from Lin’s group at Cornell, and to find an excellent and clear discussion of electroreductive carbofunctionalization of alkenes based on cathodically generated alkyl radicals. Then formulate the important and synthetically useful cathodically generated aryl radical transformations emanating from Qiu’s group. The writeup closes with a discussion of the electroreduction reaction of acyl radicals, silyl radicals, fluorosulfonyl radicals and trifluoromethoxyl radicals. The authors classified literatures into six subareas. And, the classification is unambiguous and clear. Moreover, each part has summarized the general mechanism figures and is a very readable Review for those “electro-curious” chemists who seek to rapidly explore the power of electrochemistry.

I have no doubt that the manuscript will be well received and that it will become an important part of the scientific literature for years to come. I enthusiastically recommend publication.

Minor points include:
1, Please cite a related review about cathodic reduction electrosynthesis (B. Huang, Z. Sun, G. Sun, eScience 2022, 2, 243-277.).

2, In the last outlook, the authors mentioned the paired electrolysis, please cite the following references: (1) D. Liu, Z.-R. Liu, Z.-H. Wang, C. Ma, S. Herbert, H. Schirok, T.-S. Mei, Nat. Commun. 2022, 13, 7318. (2) Z. Li, W. Sun, X. Wang, L. Li, Y. Zhang, C. Li, J. Am. Chem. Soc. 2021, 143, 3536-3543.

3, Please express “transition metal-free” or “transition-metal-free” uniformly.

4, In the line 12, please correct the “selectivity organic transformation” to “selective organic transformation”. In the line 133, “for the generations of” should be “for the generation of”. In line 443, “simultaneous” should be “Simultaneously”.

Author Response

Responses:  We thank the reviewer for the suggestions and overall positive feedback. We deeply appreciate your efforts in reviewing our manuscript. We have revised the manuscript accordingly. Our point-by-point responses are detailed below.

1: Please cite a related review about cathodic reduction electrosynthesis (B. Huang, Z. Sun, G. Sun, eScience 2022, 2, 243-277.).

Responses:  As was suggested, we have added the related review in reference 34. We thank the reviewer for the valuable comments.

2: In the last outlook, the authors mentioned the paired electrolysis, please cite the following references: (1) D. Liu, Z-R. Liu, Z.-H. Wang, C. Ma, S. Herbert, H. Schirok, T-S. Mei, Nat.Commun. 2022, 13, 7318. (2) Z. Li, W. Sun, X. Wang, L. Li, Y. Zhang, C. Li, J. Am. Chem. Soc. 2021, 143, 3536-3543

Responses:  We thank the reviewer for the comment. We have added the paired electrolysis journal article in references 106 and 107.

3: Please express"transition metal-free”or “transition-metal-free" uniformly

Responses:  We thank the reviewer for the careful correction. We have express “transition metal-free” uniformly.

4: ln the line 12, please correct the “selectivity organic transformation” to “selective organic transformation”. ln the line 133, “for the generations of” should be “for the generation of”. In line443 “simultaneous” should be “simultaneously”.

Responses:  We thank the reviewer for the careful correction. We have corrected the grammar error in manuscript, respectively.

Reviewer 2 Report

Xiang and coworkers provide a nice summary of the recent electroreductive reactions in organic synthesis. The review represents is comprehensive in properly covering many of the most significant advances to each reaction class. The figures are clear and easy to understand. The most significant criticism is in the organization of the content. The first section (2.1) is very cluttered with various examples. Subheadings would be useful for the reader, or at least some sort of table of contents that lets the reader see what content will be included. Many of the examples seems randomly selected from a large body of literature. There are no metal catalyzed reactions, despite the inclusion of radical generation mediated by organic mediators (rather than metal complexes). It’s unclear what separates these two areas or how the authors justify the distinction. Overall, this is a good review that needs some revision and reorganization to clarify its scope. 

• The entire review needs extensive copy editing to correct for grammar. It would be too much to try and highlight all of the typographical issues. 

• Confusing: “deeply positive reducing potentials of alkyl halides” followed by large negative values. Presumably means absolute values?

• Figure 2, why are all of the groups X’s but mean different things? This is unnecessarily confusing.

• P6: replace “simulated” with “stimulated”

• P6: Please clarify “lower negative” when discussing reduction potentials. 

• 51b should show a lewis acid bound “LA”, not “AL”

• The formation of compound 104 does not show a PCET pathway. Instead, it looks like an ET, PT reaction as drawn.

•For section 2.2, consider organizing reactions by mediated and unmediated electroreductive reactions.

Author Response

Responses: We deeply appreciate referee 2’s positive comments and the insightful comments, which have enabled us to improve the quality of our manuscript. We agree with the reviewer that the first section recent impetus has led to considerable advances in electroreductive reactions and we have put more effort in the presentation of this alkyl radicals parts. As was suggested, in order to be fully visible for the reader, we have added four subheadings: 2.1.1 Electroreductive Reaction of Alkyl Halides; 2.1.2 Electroreductive Reaction of Epoxides and Cyclopropanes; 2.1.3 Electroreductive Reaction of Ketones, Alkenes and Alkynes; 2.1.4 Electroreductive Reaction of Cyanoheteroarenes. We are well aware that the metal-catalyzed reductive organic electrochemical synthesis are presented in other recent reviews in Ref. 11 “Gandeepan, P.; Finger, L. H.; Meyer, T. H.; Ackermann, L., 3d metallaelectrocatalysis for resource economical syntheses. Chem. Soc. Rev. 2020, 49, 4254-4272.” and Ref. 13 “Zhu, C.; Ang, N. W. J.; Meyer, T. H.; Qiu, Y.; Ackermann, L., Organic Electrochemistry: Molecular Syntheses with Potential. ACS Cent. Sci. 2021, 7, 415-431.” or similar review that was cited in Ref. 30: Park, S. H.; Ju, M.; Ressler, A. J.; Shim, J.; Kim, H.; Lin, S., Reductive electrosynthesis: a new dawn. Aldrichimica Acta 2021, 54, 17. And Ref. 34 “Huang, B.; Sun, Z.; Sun, G., Recent progress in cathodic reduction-enabled organic electrosynthesis: Trends, challenges, and opportunities. eScience 2022, 2, 243-277.” and Ref. 35 “Zhang, S.-K.; Samanta, R. C.; Del Vecchio, A.; Ackermann, L., Evolution of High-Valent Nickela-Electrocatalyzed C−H Activation: From Cross(-Electrophile)-Couplings to Electrooxidative C−H Transformations. Chem. Eur. J. 2020, 26, 10936-10947.” To avoid overlap with these articles, but more importantly to present fundamentally new synthetic methodologies, we have decided to discuss recent successes in the area of organic electrocatalysis with a major focus on the recent trends in the use of organic redox mediators for novel electrochemical reduction findings. We have highlighted the statement in the introduction to emphasise the goal of the presented article.

1: The entire review needs extensive copy editing to correct for grammar. It would be too much to try and highlight all of the typographical issues.

Responses: We thank the reviewer for the comment. We have carefully corrected the grammar mistakes.

2: Confusing: "deeply positive reducing potentials of alkyl halides" followed by large negative values. Presumably means absolute values?

Responses: We thank the reviewer for the careful correction. We have corrected “deeply positive reducing potentials of alkyl halides” to “The high reductive potentials (absolute values) of alkyl halides”.

3: Figure 2, why are all of the groups X's but mean different things? This is unnecessarily confusing.

Responses: Thank you very much for the reviewer’s insightful comments. In Figure 2, we have corrected other two X groups to Y, Z and to show the alkyl halides, epoxides and cyclopropanes, as well as ketones and iminiums.

4:P6: replace "simulated" with "stimulated; P6: Please clarify “lower negative" when discussing reduction potentials

Responses: We thank the reviewer for the careful correction. We have corrected “simulated” to “stimulated” in manuscript. As was suggested, when discussing reductive potentials, we have explained the absolute value in parentheses.

5:51b should show a lewis acid bound “LA” not “AL”

Responses: We thank the reviewer for the careful correction. We have corrected the “LA” in 40b compound.

6:The formation of compound 104 does not show a PCET pathway. Instead, it looks like an ET, PT reaction as drawn.

Responses:  We thank the reviewer for the comment. In Figure 11B, we have corrected the formation of compound 104 (the new No. is 103) by a PCET pathway and in Figure 11C, we have corrected the formation of compound 106 by an ET pathway.  

7:For section 2.2, consider organizing reactions by mediated and unmediated electroreductive reactions.

Responses:  We thank the reviewer for the suggestion and the nice assessment for an alternative structure of the section 2.2. We have organized the aryl radicals parts by direct and indirect electroreduction.

Round 2

Reviewer 2 Report

Thank you to the authors for the corrections and changes. I support publication of the revised version of this Review.